Reliability and validity of the multi-point method and the 2-point method’s variations of estimating the one-repetition maximum for deadlift and back squat exercises

Çetin Onat 1
Akyildiz Zeki 2
Demirtaş Barbaros 3
Sungur Yılmaz 4 5
http://orcid.org/0000-0001-9813-2842 Clemente Filipe Manuel 6 7 8
Cazan Florin 9
http://orcid.org/0000-0001-7677-5070 Ardigò Luca Paolo 10 luca.ardigo@univr.it
1 Faculty of Sports Sciences, Department of Coaching Education, Yalova University , Yalova , Turkey
2 Faculty of Sports Sciences, Gazi University , Ankara , Turkey
3 Movement and Training Sciences Department, Sports Sciences Faculty, Sakarya Applied Sciences University , Sakarya , Turkey
4 Department of Movement and Training Science, Faculty of Sports Sciences, Akdeniz University , Antalya , Turkey
5 Sports Medicine and Athletic Performance Department, Gloria Sports Arena , Antalya , Turkey
6 Escola Superior Desporto e Lazer, Instituto Politécnico de Viana do Castelo , Viana do Castelo , Portugal
7 Research Center in Sports Performance, Recreation , Innovation and Technology (SPRINT) , Melgaço, Portugal
8 Instituto de Telecomunicações, Delegação da Covilhã , Lisboa, Portugal
9 Faculty of Physical Education and Sport, Ovidius University of Constanta , Constanta , Romania
10 Department of Neurosciences, Biomedicine and Movement Sciences, School of Exercise and Sport Science, University of Verona , Verona , Italy
Cè Emiliano
Electronic publication date: 2022 Mar 2
Publication date: 2022
Volume: 10
Electronic Location ID: e13013
Received 2021 Oct 18; Accepted 2022 Feb 4
Copyright: © 2022 Çetin et al.
Copyright year: 2022
Copyright holder: Çetin et al.
License: This is an open access article distributed under the terms of the Creative Commons Attribution License, which permits unrestricted use, distribution, reproduction and adaptation in any medium and for any purpose provided that it is properly attributed. For attribution, the original author(s), title, publication source (PeerJ) and either DOI or URL of the article must be cited.
License URL: https://creativecommons.org/licenses/by/4.0/

Keywords: Load-velocity relationship, Multipoint method, Two-point method, Velocity-based training

Funding: Fundação para a Ciência e Tecnologia/Ministério da Ciência, Tecnologia e Ensino Superior through national funds and co-funded with EU funds under the project UIDB/50008/2020 FMC was funded by Fundação para a Ciência e Tecnologia/Ministério da Ciência, Tecnologia e Ensino Superior through national funds and co-funded with EU funds under the project UIDB/50008/2020. The funders had no role in study design, data collection and analysis, decision to publish, or preparation of the manuscript.

==============================
This study aimed at examining the concurrent validity and reliability of the multi-point method and the two-point method’s variations for estimating the one-repetition maximum (1RM) in the deadlift and squat exercises and to determine the accuracy of which optimal two loads can be used for the two-point method protocol. Thirteen resistance-trained men performed six sessions that consisted of two incremental loading tests (multi-point method: 20–40–60–80–90% and two-point method variations: 40–60%, 40–80%, 40–90%,60–80%, 60–90%) followed by 1RM tests. Both the multi-point method and the two-point method load variations showed reliable results for 1RM estimation (CV < 10%) squat and deadlift exercises. Session-session reliability was found to be low in deadlift (ICC: 0.171–0.335) and squat exercises (ICC: 0.235–0.479) of 40–60% and 60–80% in two-point methods. Deadlift (ICC: 0.815–0.996) and squat (ICC: 0.817–0.988) had high session-to-session reliability in all other methods. Regarding the validity of deadlift exercise, the multipoint method (R2 = 0.864) and two variations of the two-point method (R2 = 0.816 for 40–80%, R2 = 0.732 for 60–80%) showed very large correlations, whereas other two variations of the two-point method (R2 = 0.945 for 40–90%, R2 = 0.914 for 60–90%) showed almost perfect correlations with the actual 1RM. Regarding the validity of squat exercise, the multi-point method (R2 = 0.773) and two variations of the two-point method (R2 = 0.0847 for 60–80%, R2 = 0.705 for 40–90%) showed very large correlations, whereas 40–60% variation showed almost perfect correlation (R2 = 0.962) with the actual 1RM. In conclusion, whereas both the multi-point method and the two-point method load variations showed reliable results, the multiple-point method and most of the two-point methods’ load variations examined in this research provided an accurate (from large-moderate to perfect) estimate of the 1RM. Therefore, we recommend using the multi-point method and especially the two-point methods variations including higher relative loads to estimate 1RM.

Introduction

Continuous improvement of strength and power variables is required to increase athletic performance (Suchomel et al., 2018). Exercise intensity or load is generally acknowledged as the most important stimulus related to changes in strength levels and has been commonly identified as a percentage of maximum strength (Crewther et al., 2006; González-Badillo & Sánchez-Medina, 2010). Maximal strength is related to improved force-time characteristics and general and sport-specific skill performance. Therefore, it plays an essential role in most sports (Wisløff et al., 2004). The one-repetition maximum (1RM) is widely used in practice and scientific research to determine training intensities and loads during the resistance training programs (Wood, Maddalozzo & Harter, 2002; Kraemer & Ratamess, 2004; Scott et al., 2016). The 1RM can be determined either directly or indirectly, but the direct determination of the 1RM from one maximal lift has several disadvantages, such as an increase in the risk of injury if performed incorrectly or by inexperienced subjects (Balsalobre-Fernández et al., 2018; Ruf, Chéry & Taylor, 2018). Besides, it is time-consuming and possibly impractical for large groups and requires fully motivated subjects to achieve their actual 1RM (Niewiadomski et al., 2008; Bazuelo-Ruiz et al., 2015). In addition, individual stress factors (e.g., sports stress, life stress, and social stress) cause fluctuations in the ability to move an external resistance (Fry & Kraemer, 1997), and this can affect the daily stability of the 1RM level (Mann, Ivey & Sayers, 2015). Therefore, considering the disadvantages of 1RM tests, researchers have developed various indirect prediction methodologies and regression equations to estimate 1RM using submaximal loads (Kravitz et al., 2003).

In recent years, among the indirect methodologies to estimate 1RM, the most widely used is the load-velocity (L-V) relationship (Jidovtseff et al., 2011; González-Badillo & Sánchez-Medina, 2010). Several researchers have reported a highly linear relationship between movement velocity and relative load (%1RM) in various resistance training exercises due to advancements in kinetic and kinematic transducer technologies that accurately measure bar velocity (González-Badillo & Sánchez-Medina, 2010; Sánchez-Medina & González-Badillo, 2011; Sánchez-Medina et al., 2014; Conceição et al., 2016). The L-V relationship equations have been used to predict the 1RM by using submaximal loads (Jidovtseff et al., 2011; García-Ramos et al., 2018b). The general load-velocity (L-V) relationships (Hecksteden et al., 2018) and the individual L-V relationships that use regression equations as the load associated with the mean velocity (MV) of the 1RM were proposed to estimate the 1RM (Weakley et al., 2021). Although the researchers reported a strong and approximately linear L-V relationship for the general method (González-Badillo & Sánchez-Medina, 2010; Sánchez-Medina & González-Badillo, 2011; Conceição et al., 2016; Muñoz-López et al., 2017; García-Ramos et al., 2018b), the affecting of MV and L-V relationship by the type (Sánchez-Medina & González-Badillo, 2011; Helms et al., 2017) and implementation of exercises (García-Ramos et al., 2018b; Pérez-Castilla et al., 2020) and sample groups (Torrejón et al., 2019) may limit its use in practice. As an alternative to generalized group equations, the multiple-point method applied to the individual L-V relationship was proposed for a more accurate estimate of the value of 1RM (Jovanović & Flanagan, 2014). The individual load-velocity relationship consists of evaluating the velocity outputs of multiple submaximal loads ranging from 4 or 9 (Helms et al., 2017). The multipoint method also showed an accurate estimate of the 1RM value (Jidovtseff et al., 2011; Picerno et al., 2016; Helms et al., 2017; Pérez-Castilla et al., 2020), but multi-point-based individualized L-V protocols can often be time-consuming and prone to fatigue (García-Ramos et al., 2018a).

Considering the limitations of the two methods mentioned above and assuming that the L-V relationship in resistance exercises is stable and linear (González-Badillo & Sánchez-Medina, 2010; Bazuelo-Ruiz et al., 2015; Conceição et al., 2016), the two-point method has been proposed by Jaric as an alternative (Jaric, 2016). The two-point method analyzes two different external loads to determine the L-V relationship instead of multiple loads (McBurnie et al., 2019). The two-point method has a quick and fatigue-free procedure to predict the 1RM (Weakley et al., 2021). The researchers recommend using two distant loads corresponding to approximately 45%, 50–80%, 85% of 1RM in the protocol of the two-point method (García-Ramos et al., 2018a; Pérez-Castilla et al., 2021). However, the question of what should be the optimal two loads to be used in this method is still current (García-Ramos et al., 2018a; Pérez-Castilla et al., 2018).

The validity of the two-point method for upper-body free-weight exercises such as bench pull (García-Ramos et al., 2019), bench press (Jiménez-Alonso et al., 2020), lat pull-down, and seated cable row (Pérez-Castilla et al., 2021) has been confirmed. However, its validity for free-weight lower-body exercises (e.g., conventional deadlift, squat) has never been explored, and the researchers assume that the inconsistencies in the prediction accuracy of lower body exercises may be due to the technical complexity (Weakley et al., 2021) of these movements. This hypothesis needs to be scientifically proven by standardizing the application techniques of the exercises during the research. For the reasons stated above, the aims of this study were (1) to examine the concurrent validity and reliability of the multi-point method and the two-point method’s variations for estimating 1RM in the deadlift and squat exercises and (2) to identify the optimal combination of the two loads that allows predicting the 1RM with the highest accuracy.

Materials and Methods

Study design

In this study, we investigated the reliability and validity of the multiple-point method and two-point method to predict 1RM for back squat and deadlift. The linear regression equation was computed using an individualized load-velocity relationship using mean velocity data of different submaximal loads (40%, 60%, 80%, 90%). For the multiple-point method, each submaximal load and velocity were considered, meanwhile different two-point combinations (40–60%, 40–80%, 40–90%, 60–80%, 60–90%) were used to predict 1RM.

Investigations had been conducted for 6 days separated by 48 h resting periods. On the first day and the fourth day, subjects’ squat 1RM and the deadlift 1RM loads were determined and familiarizations were done for further velocity-based assessments. On the second, third, fifth and sixth day, 1RM tests were conducted while saving barbell velocity data to predict 1RM by using a linear regression equation.

Participants

Thirteen resistance-trained men participated in the study (23.6 ± 4.2 years, 179.6 ± 7.1 m, 80.2 ± 8.9 kg). Including criteria were performing both exercises with a good pattern, at least 2 years of strength training experience. All subjects were free from any musculoskeletal injuries. Before the study, all subjects were informed of the purpose and risks of the investigation and provided signed informed consent. This study was approved by Yalova University Clinic Researches Ethics Committee (protocol number 2021/13, 11/03/2021) and was conducted in accordance with the Declaration of Helsinki.

Procedures

Athletes performed six testing sessions separated by 48 h each other. Squat assessments were done on the first, the second, and the third day, whereas deadlift assessments were done on the fourth, the fifth, and the sixth day. On the first day, and the fourth day, familiarization sessions were conducted for each exercise. In these sessions, self-determined 1RM’s relative loads were used to determine the actual 1RM score. Subjects began with a standardized warm-up followed by an incremental 1RM test. On the second and the third day for squat, on the fifth and the sixth day for deadlift subjects followed the same warm-up protocol and 1RM assessments. Relative loads that were used in the second, the third, the fifth, and the sixth day were determined from the initial 1RM assessment.

Regarding 1RM assessment, before each test, a standardized warm-up consisting of 8 min of jogging, 5 min of lower extremity mobility exercises, and 3 reps of the deadlift and the squat with 20 kg barbell was done. A custom-built power cage (Life Fitness, Hammer Strength, Rosemont, IL, USA) and 20 kg barbell (Eleiko, Halmstad, Sweden) were used for all 1RM assessments. In the first familiarization session, subjects were taught to perform squat at 90° of the knee joint for the standardization of squat exercise. A string placed between two tripods, set for each individual to make them perform squat at 90° was used for all squat assessments. Each athlete touched the string with their hip in the eccentric phase of squat movement (Bazuelo-Ruiz et al., 2015). When the eccentric phase of the squat was ended at 90°, subjects waited for 2 s to inactivate the stretch-shortening cycle and performed the concentric phase as fast as possible. On the fourth day (familiarization day of deadlift assessments), a technical model was shown and explained for deadlift exercise. Rounded lower or upper back and no full extension of the hip were taken into account as a failure (Ruf, Chéry & Taylor, 2018). For the beginning and the finishing position of the deadlift, weight plates hat to touch the ground. When the eccentric phase of deadlift was ended, subjects waited for 2 s to inactivate the stretch-shortening cycle and performed the concentric phase as fast as possible. Verbal encouragement was given for each exercise. Incremental test protocol was adapted from previous research (Banyard, Nosaka & Haff, 2017). For the familiarization sessions, subjects started with 20% loads of their self-determined 1 repetition maximum. They performed 3 reps at 20% 1RM, 3 reps at 40% of 1RM, 3 reps at 60% of 1RM, 1 rep at 80% of 1RM, and 1 rep at 90%. The load was increased 2.5–5 kg after each successful attempt until the subject failure to lift. The rest period between sets was 3 min. For the velocity assessments, relative loads of 1RM that were determined in the 1RM test were used. Subjects performed the same warm-up and the same incremental test protocol.

Regarding velocity measurements, barbell velocity data was collected by means of a linear position transducer GymAware (GymAware Power Tool; Kinetic Performance Technologies, Canberra, Australia), whose validity was reported in previous studies (O’Donnell et al., 2018; Thompson et al., 2020). For each relative load, the highest mean velocity (MV) data was saved for statistical analysis. The device was connected to a tablet (iPad; Apple Inc., CA, USA) through bluetooth and barbell velocity data was recorded with the GymAware application (software version 1.2.1).

Statistical analyses

All the statistical procedures were computed in an Excel spreadsheet designed by Hopkins (2015). Descriptive statistics were calculated for all variables and reported as mean ± standard deviations (SD). The normality of the data was examined using the Kolmogorov–Smirnov test. It was determined that the data showed normal distribution. The level of the relationship between the measurements obtained with different load velocity profile methods was determined by means of the Pearson correlation test. The results of different methods using linear regression analysis were transformed into images using a correlation graph and r values and confidence intervals were drawn on the graphs. One-way ANOVA test was used to examine the differences of estimates obtained by the different load-velocity methods. Inter-session reliability was expressed with coefficients of variation (CV) calculated from mean and standard deviations (standard error of measurement (typical error)/participants mean score × 100). The uncertainty was expressed as 95% confidence interval (CI). Smallest worthwhile change (SWC) was calculated by multiplying the between-subject standard deviation (SD) of the performance by 0.2 (SWC = 0.2 between-subject SD [42], Lacome et al., 2019). The validity of different load velocity estimation methods was tested using the Bland-Altman plot systematic deviation±random error. The presence of heteroscedasticity was analyzed using correlation (R2) between absolute differences for validity.

As the level of the relationship, minor (0.0), small (0.1), medium (0.3), large (0.5), very large (0.7), almost perfect (0.9) and perfect (1.0) levels were used (Hopkins, 2015). The CV was rated as good when CV < 5%, as moderate when CV was 5–10%, and as poor when CV was >10%. p < 0.05 alpha level was used as the significance level.

Results

Table 1 shows the difference between the estimates obtained by means of the different methods in the deadlift exercise and the values obtained with the reference (“real”) method. The inter-session reliability of the methods used in the deadlift exercise in two separate sessions is shown in Table 2. Table 3 shows the difference between the estimates obtained by different methods in the squat exercise and the values obtained with the real method. The inter-session reliability of the methods used in the squat exercise in two separate sessions is shown in Table 4.

Table 1 Differences between methods of data derived from the load velocity profile in deadlift exercise.

Tukey’s multiple comparisons test	Mean difference	95% CI of difference	p value	Mean absolute error	
Deadlift real RM vs Deadlift multiple method	−3.16	[−48.13 to 41.81]	>0.9999	7.057	
Deadlift real RM vs Deadlift 40–60%	−47.59	[−92.56 to −2.62]	0.0305*	52.583	
Deadlift real RM vs Deadlift 40–80%	−12.28	[−57.24 to 32.69]	0.9830	12.911	
Deadlift real RM vs Deadlift 40–90%	−2.995	[−47.96 to 41.97]	>0.9999	5.556	
Deadlift real RM vs Deadlift 60–80%	−8.23	[−53.20 to 36.74]	0.9981	11.775	
Deadlift real RM vs Deadlift 60–90%	−2.97	[−47.93 to 42.00]	>0.9999	6.78	
Note:

* p < 0.05.

Table 2 Inter-session reliability of the data obtained from the load-velocity profile in the deadlift exercise.

Measurement methods	CV%	SWC	ICC	
Deadlift multiple method	4.93	4.73	0.996	
Deadlift 40–60%	7.72	7.38	0.171	
Deadlift 40–80%	4.23	3.40	0.815	
Deadlift 40–90%	4.27	3.20	0.996	
Deadlift 60–80%	3.86	2.65	0.335	
Deadlift 60–90%	4.62	3.53	0.972	

Table 3 Differences between methods of data derived from the load velocity profile in squat exercise.

Tukey’s multiple comparisons test	Mean difference	95% CI of difference	p value	Mean absolute error	
Squat real RM vs Squat multiple	−14.95	[−73.43 to 43.53]	0.9879	20.211	
Squat real RM vs Squat 40–60%	−35.12	[−93.60 to 23.36]	0.5535	47.075	
Squat real RM vs Squat 40–80%	−23.70	[−82.18 to 34.78]	0.8889	33.335	
Squat real RM vs Squat 40–90%	−9.223	[–67.70 to 49.26]	0.9992	18.668	
Squat real RM vs Squat 60–80%	−43.58	[−102.1 to 14.90]	0.2874	61.294	
Squat real RM vs Squat 60–90%	−7.31	[–65.79 to 51.17]	0.9998	16.693	

Table 4 Inter-session reliability of the data obtained from the load-velocity profile in the squat exercise.

Measurement methods	CV%	SWC	ICC	
Squat multiple method	4.42	4.72	0.817	
Squat 40–60%	6.26	6.66	0.235	
Squat 40–80%	4.07	4.40	0.822	
Squat 40–90%	4.34	4.60	0.905	
Squat 60–80%	9.81	10.25	0.479	
Squat 60–90%	4.50	4.74	0.988	

Figures 1 and 2 show the differences of different methods from the real value. Figures 3 and 4 show the relationships of different methods with the real value. Bland-Altman plots showing the differences between data methods derived from the load velocity profile in exercises are shown in Figs. 5 and 6. Both the multi-point method and the two-point method load variations showed reliable results for 1RM estimation (CV < 10%) squat and deadlift movements. Session-session reliability was found to be low in deadlift (ICC: 0.171–0.335) and squat exercises (ICC: 0.235–0.479) of 40–60% and 60–80% in two-point methods. Deadlift (ICC: 0.815–0.996) and squat (ICC: 0.817–0.988) had high session-to-session reliability in all other methods. Regarding the validity of deadlift exercise, the multipoint method (R2 = 0.864) and 2 variations of two-point method (R2 = 0.816 for 40–80%, R2 = 0.732 for 60–80%) showed very large correlations, whereas other two variations of the two-point method (R2 = 0.945 for 40–90%, R2 = 0.914 for 60–90%) showed almost perfect correlations with the actual 1RM. Regarding the validity of squat exercise, the multi-point method (R2 = 0.773) and two variations of the two-point method (R2 = 0.0847 for 60–80%, R2 = 0.705 for 40–90%) showed very large correlations, whereas 40-60% variation showed almost perfect correlation (R2 = 0.962) with the actual 1RM.

Figure 1 Differences between methods of data derived from the load velocity profile in deadlift exercise.

Figure 2 Differences between methods of data derived from the load velocity profile in squat exercise.

Figure 3 Relationships of data derived from the load velocity profile in deadlift exercise between methods.

Figure 4 Relationships of data derived from the load velocity profile in squat exercise between methods.

Figure 5 Bland-Altman plot showing the differences between methods of data derived from the load velocity profile in deadlift exercise.

Figure 6 Bland-Altman plot showing the differences between methods of data derived from the load velocity profile in squat exercise.

Discussion

This study aimed at examining the concurrent validity and reliability of the multi-point method and the two-point method for estimating 1RM in the deadlift and squat exercises and to determine the accuracy of which optimal two loads variation can be used for the two-point method protocol. Our main findings revealed that both the multi-point method and the two-point method variations showed reliable results for 1RM estimation (CV < 10%) squat and deadlift exercises. Session-session reliability was found to be low in deadlift (ICC: 0.171–0.335) and squat exercises (ICC: 0.235–0.479) of 40–60% and 60–80% in two-point methods. Deadlift (ICC: 0.815–0.996) and squat (ICC: 0.817–0.988) had high session-to-session reliability in all other methods.

Regarding the validity of deadlift exercise, the multipoint method (R2 = 0.864) and two variations of two-point method (R2 = 0.816 for 40–80%, R2 = 0.732 for 60–80%) showed very large correlation, whereas other two variations of the two-point method (R2 = 0.945 for 40–90%, R2 = 0.914 for 60–90%) showed almost perfect correlation with the actual 1RM. Regarding the validity of squat exercise, the multi-point method (R2 = 0.773) and two variations of the two-point method (R2 = 0.0847 for 60–80%, R2 = 0.705 for 40–90%) showed very large correlation, whereas 40-60% variation showed almost perfect correlation (R2 = 0.962) with the actual 1RM.

Blant-altman plots show that the differences between the real RM values of the multi-point and two-point method’s variations (40–80%, 40–90%, and 60–80%) in deadlift exercise are between acceptable upper and lower limits. However, it shows that the 40–60% and 60–90% variations are not between the acceptable upper and lower limits with real RM percentages. Details of the results are given in Fig. 5. For the squat exercise, Blant-altman plots show that the differences between the real RM values of the multi-point and two-point method’s variations (40–80%, 40–90%, and 60–90%) are between acceptable upper and lower limits. However, it shows that the 40–60% and 60–80% variations are not between the acceptable upper and lower limits with real RM percentages. Details of the results are given in Fig. 6.

Past studies that examined 1RM estimation methods using the L-V relationship reported different results for lower-body (squat, deadlift) and upper-body (bench press) compound exercises. For example, whereas the general validity results of studies on bench press exercise (González-Badillo & Sánchez-Medina, 2010; Jidovtseff et al., 2011; Sánchez-Medina et al., 2014; Loturco et al., 2017; García-Ramos et al., 2018a) report a high correlation for L-V relationship and 1RM, the results for squat and deadlift exercises seem to be inconsistent (Sánchez-Medina & González-Badillo, 2011; Bazuelo-Ruiz et al., 2015; Banyard, Nosaka & Haff, 2017; Lake et al., 2017). Researchers explain this result variation in lower-body exercises for several reasons. Although we attach importance to standardization practices among the measurements in the present study, we especially associate the differentiation of reliability and validity results with the following reasons. Weakley et al. (2021) attribute discrepancies in the accuracy of 1RM estimation to the greater technical complexity of squat and deadlift exercises compared with upper-body exercises. On the other hand, Balsalobre-Fernández & Torres-Ronda (2021) indicate that oscillations cause angle deviations up to 20° in exercises that require vertical lifting such as squat. This situation may cause measurement errors in studies where free weight exercises with angular momentum components are examined with linear transducers, as in this study. Therefore, they stated that most of the studies analyzing the L-V relationship use the Smith Machine tool, which guarantees the vertical pathway of the barbell to avoid horizontal displacements. In fact, the researchers tried to use multiple linear position transducers instruments placed in four directions (positioned above and anterior, above and posterior) in order to avoid asymmetrical differences in the barbell in these exercises (Banyard, Nosaka & Haff, 2017).

Past studies that examined the L-V relationships different methods in free weight lower body exercises to estimate 1RM also reported variable results on validity and reliability. Banyard, Nosaka & Haff (2017) found that the variations of the multiple-point method for back squat exercise differ from three to five loads and they stated that the L-V relationship was moderately reliable and valid, but could not accurately predict the 1RM. Yet, in extensive reviews written on 1RM estimation methods, the authors concluded that the multi-point method can be time-consuming and prone to fatigue (Garcia-Ramos & Jaric, 2018; McBurnie et al., 2019). Although Ruf, Chéry & Taylor (2018) reached reliable results in their research in which they examined various variations of the two-point method in deadlift exercise, they reported that predicted 1RM scores computed from all submaximal load ranges substantially overestimated the actual 1RM with considerable differences between athletes. Lake et al. (2017) found more extreme results compared with the studies highlighted above. Whereas they reported poor to moderate reliability, they found that the estimated deadlift 1RMs were as much as 15% less than actual 1RMs. For the estimation methods used in lower-body exercises that cannot accurately measure the actual 1RM, the researchers also recommend acceptable kg errors. For example, Jukic et al. (2020) compared six prediction methods, including the multi-point and two-point methods (40–90%), and found that no method for deadlift exercise could accurately predict the actual 1RM. That is why they have suggested error limits of 5 kg to sports professionals. In addition to biomechanical and exercise technique factors in lower body exercises, the fact that minimal velocity thresholds (MVT) are exercise-specific should also be considered. Whereas researchers have found back squat MVT to be around 0.30 ms−1, others have recorded deadlift 1RM velocities as 0.16–0.17 ms−1 (Izquierdo et al., 2006; Sanchez-Medina, Perez & Gonzalez-Badillo, 2010). So that in the regression analysis we used 0.30 ms−1 for squat and 0.16 ms−1 for deadlift as previous researches reported.

The two-point method is more practical and requires less time than the multipoint method, but there is no strong scientific opinion about which two most appropriate loads should be preferred with this method. García-Ramos et al. (2018a) recommended different external loads representing approximately 50% and 80% of self-reported 1RM. Based on our experience during the measurements of this research, we recommend not using loads (multi-point method) below 40% for light loads due to velocity inconsistencies. The results of our research show that the other load, which is heavier, should be around 90%. Although this situation negatively affects the advantages of the two-point method on the formation of fatigue, its validity decreases as this load rate decreases.

According to the validity results of this study, it was determined that the validity values of the two-point method variations were higher than the multiple-point method. These results further strengthened the advantages of the two-point method, such as requiring less time and being more practical. However, the problem of which two loads would be most appropriate to estimate 1RM in the two-point method remains. In our study, two load variations with high validity in squat and deadlift exercises were found to be different. We think that this is because the velocity thresholds are exercise-specific. Whereas García-Ramos et al. (2018a) suggest using two very different loads in this method, researchers also stated that heavy relative loads in the two-point method may cause problems in inexperienced athletes. Most of the variations with high validity in this study contain loads of 80% or more. Therefore, we think that the experience and strength level of the athlete can be an important criterion in the use of the two-point method.

To the best of our knowledge, this is the only study that examined the reliability and validity of the multi-point and two-point 1RM estimation methods in both squat and deadlift exercises. The two-point method has never been validated during lower-body exercises yet (Weakley et al., 2021). However, the results of this research show that valid results can be obtained in many versions of the two-point method if the measurement and standardization problems in lower body exercises are solved. Despite all these estimation methods, researchers state that direct measurement of 1RM in lower body exercises is more reliable than L-V relationship methods using submaximal loads (García-Ramos et al., 2018a).

Conclusions

In conclusion, whereas both the multiple-point method and two-point method loads’ variations showed reliable results, the multiple-point method and most of the two-point method load variations examined in this research provided (from large moderate to perfect) accurate estimate of the 1RM. Therefore, we recommend using the multi-point method and especially two-point methods variations including higher relative loads in estimating 1RM. While the two-point method may overcome some of the potential limitations associated with the direct determination of 1RM strength, professionals should also be aware that direct determination of 1RM appears to be the most reliable method in training programs where daily 1RM level is not essential. Lastly, strength and conditioning practitioners and sports scientists should take into account that MV was used for this research and other velocity types like MPV and PV may give different results.

Supplemental Information

Supplemental Information 1 Raw Data.

Click here for additional data file.

We thank all subjects, who participated in the study.

Additional Information and Declarations

Competing Interests

Author Contributions

Human Ethics

Data Availability

Filipe M. Clemente & Luca P. Ardigò are Academic Editors for PeerJ.

Onat Çetin conceived and designed the experiments, performed the experiments, analyzed the data, prepared figures and/or tables, authored or reviewed drafts of the paper, and approved the final draft.

Zeki Akyildiz conceived and designed the experiments, performed the experiments, analyzed the data, prepared figures and/or tables, authored or reviewed drafts of the paper, and approved the final draft.

Barbaros Demirtaş conceived and designed the experiments, performed the experiments, analyzed the data, prepared figures and/or tables, authored or reviewed drafts of the paper, and approved the final draft.

Yılmaz Sungur conceived and designed the experiments, performed the experiments, analyzed the data, prepared figures and/or tables, authored or reviewed drafts of the paper, and approved the final draft.

Filipe Manuel Clemente conceived and designed the experiments, performed the experiments, analyzed the data, prepared figures and/or tables, authored or reviewed drafts of the paper, and approved the final draft.

Florin Cazan conceived and designed the experiments, performed the experiments, analyzed the data, prepared figures and/or tables, authored or reviewed drafts of the paper, and approved the final draft.

Luca Paolo Ardigò conceived and designed the experiments, performed the experiments, analyzed the data, prepared figures and/or tables, authored or reviewed drafts of the paper, and approved the final draft.

The following information was supplied relating to ethical approvals (i.e., approving body and any reference numbers):

This study was approved by Yalova University Clinic Researches Ethics Committee (protocol number 2021/13, 11/03/2021) and was conducted in accordance with the Declaration of Helsinki.

The following information was supplied regarding data availability:

The raw measurements are available in the Supplemental File.

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
