# Peer review of "Reliability and validity of the multi-point method and the 2-point method’s variations of estimating the one-repetition maximum for deadlift and back squat exercises"

_PeerJ, doi:10.7717/peerj.13013_

## Round 0.1 · original submission · Major Revisions

Two experts revised your ms and retrieved several major flaws you should consider in the further submission.

Reviewer 1 ·

Basic reporting

This study aimed to examine the concurrent validity and reliability of the multi-point method and the two-point method’s variations for estimating the one-repetition maximum (1RM) in two exercises: the deadlift and squat exercises. Participants performed six sessions of 2 incremental loading tests: multi-point method and two-point method variations, followed by 1RM tests. The authors conclude that the multiple-point method and most of the two-point method load variations examined in this research provided (from large moderate to perfect) an accurate estimate of the 1RM. Therefore, they recommend using the multi-point method and specially two-point methods variations, including higher relative loads in estimating 1RM.
The introduction of the manuscript is sufficient, and appropriate bibliographical references are included.
Figures and tables can be considered adequate and clear.

Experimental design

Numerous studies have been done lately to estimate the value of 1RM, but, to our knowledge, in this case, a set of predictors not previously used are applied.

The methodology is well described. However, the procedure for determining the reliability of the measures should include the intraclass correlation coefficient (ICC) and the coefficient of variation (CV) based on the total root mean square within-subjects of the repeated measures ANOVA analysis.

On the other hand, the mean velocity (MV), not the mean propulsive velocity (MPV), is used for the analysis of the load-velocity relationship. It is known that at low percentages, the MPV gives slightly higher values than the MV. This circumstance may result in the best fit of the percentage-velocity relationship not being a straight line but a curve. Although the authors conclude that high loads should be used as a reference to estimate 1RM, the results could be different if the MPV is used.

When the percentage values used for prediction are very different, a major problem can occur. For example, when the subject jumps from 60 to 90% of 1RM, he may not be in the best condition to obtain his maximum performance at 90% since he has not received sufficient prior stimulus to move such a high load. Likewise, it is unlikely to perform to the best of the subject's ability at 60% without prior warm-up. If any lifting is performed prior to 60% and some intermediate between the two percentages, it should be indicated in the methodology. This warm-up is necessary to arrive in the best state of potentiation to the high loads. The best example is the progressive increase of loads that weightlifters perform in training and competition. These previous lifts do not have a negative effect by fatigue but a positive effect by potentiation.

In the discussion section, we have not seen reference to the analyses with the Bland-Altman plots.

Validity of the findings

The following reasons could condition the validity of the results:
1. The CV values are very high in both exercises. It does not seem reasonable to recommend the application of a procedure that does not present sufficient reliability.
2. If a warm-up has not been used prior to the lower of the percentages and between the two percentages, likely, the performances were not the maximum that the participants could achieve.
3, As the authors themselves indicate, the value of 1RM can change frequently. Therefore, the recommendation to use any of the procedures indicated would require performing a 1RM test every day or at least very frequently since the absolute loads representing the different percentages could have changed.
4. It should be explained why, for example, the correlation obtained with 40-90% is 0.839, 60-80% is 0.062, and 60-90% is 0.872. The study suggests that the higher the loads, the more accurate the prediction. Therefore, the relationship obtained with 60-80% should be at least similar to the other two indicated

Reviewer 2 ·

Basic reporting

Reliability analysis should be improved.

Experimental design

Excellent!

Validity of the findings

They are OK, but reliability analysis should be performed again and absolute errors in 1RM prediction shouls also be provided.

Additional comments

Authors should be congratulated for a general well-written article and the examination of two basic exercises less investigated in the VBT literature. The article is interesting but I have two main considerations: (I) I believe that reliability was not properly evaluated (see comment below) and therefore all the discussion related to the reliability indices should be modified after doing the proper calculation, and (II) absolute errors should be added to the validity analysis. Some specific comments can be found below:


Line 34. Replace “consisted in” by “consisted of”.

Line 115-116. Replace to: “to identify the optimal combination of the two loads that allows predicting the 1RM with the highest accuracy”.

Method is OK but I think it would be clearer if divided in different subsections (participants, study design, procedures, and statistical analyses).

Line 186. It is not clear how the CV was calculated for reliability analyses. CV should be calculated as the standard error of measurement (typical error) / participants´mean score x 100). I believe the CV value provided in this version of the manuscript depicts the between-subject variability instead of the within-subjects variability, and this is the reason why authors found a poor reliability but in fact I think that the reliability has not been analyses yet because the current computation of the CV is not appropriate.

Table 1 and 3. authors should also provide the absolute errors.

Table 2 and 4. The huge CV also for the real RM is very strange. This confirms that reliability has not been calculated properly.

Line 262. Which minimal velocity threshold did you use?

Line 271. García-Ramos and Perez-Castilla did not recommend loads below for 1RM prediction. They have recommended loads below 40%1RM for assessing the F-V relationship which are different things. The differences between both relationships should be better explained here, and clarify that the 2 optimal loads depend on the relationship to be determined: F-V requires a point close to v0 and another close to F0, and L-V the lower point always recommended by García-Ramos and Perez-Castilla is approximately 50%1RM and the higher should be closer to 1RM (aprox. 85%1RM). The rest of the arguments I think they are quite solid!

Line 293. And this can be checked with your data, which is very interesting, when you do the proper reliability analysis!

---

## Round 0.2 · Major Revisions

Dear Authors,

The reviewer was not able to see some of the changes made in the previous revision. Please check the correctness of the procedure used to upload the revised files.

Reviewer 2 ·

Basic reporting

Dear authors, thank you for your compliments regarding the usefulness of my previous review. My main two points was the proper calculation of CV values and the calculation of the absolute errors for determining the accuracy in the estimation of the 1RM. Authors indicated that new CV values were added to tables 2 and 4, but I do not see any chanhe implemented in these tables. Results in the text are changed but in the tables are the same. Now the results presented in the table do not match the results presented in the Tables. In addition, authors indicated that they added the absolute errors in the estimation of the 1RM for the different models, but I was not able to see these results in the revised version of the manuscript.

It is very important to me to implement these changes clearly in the revised version of the manuscript.

Experimental design

No comment

Validity of the findings

No comment

Additional comments

No comment

---

## Round 0.3 · accepted · Accept

Dear Authors,

The reviewer read the revision of your last manuscript version and found your changes adequate.